# Accessory Genomic Epidemiology of Cocirculating *Acinetobacter baumannii* Clones

Valeria Mateo-Estrada,[a] José Luis Fernández-Vázquez,[b] Julia Moreno-Manjón,[b,d] Ismael L. Hernández-González,[a] Eduardo Rodríguez-Noriega,[c] Rayo Morfín-Otero,[c] María Dolores Alcántar-Curiel,[b] Santiago Castillo-Ramírez[a]

[a]Programa de Genómica Evolutiva, Centro de Ciencias Genómicas, Universidad Nacional Autónoma de México, Cuernavaca, México

[b]Laboratorio de Infectología, Microbiología e Inmunología Clínica, Unidad de Investigación en Medicina Experimental, Facultad de Medicina, Universidad Nacional Autónoma de México, Ciudad de México, México

[c]Hospital Civil de Guadalajara Fray Antonio Alcalde e Instituto de Patología Infecciosa y Experimental, Centro Universitario de Ciencias de la Salud, Universidad de Guadalajara, Guadalajara, Jalisco, México

[d]Laboratorio de Bacteriología Médica, Posgrado en Ciencias Quimicobiológicas, Escuela Nacional de Ciencias Biológicas, Instituto Politécnico Nacional, Carpio y Plan de Ayala SN, Ciudad de México, México

**ABSTRACT** *Acinetobacter baumannii* has become one of the most important multidrug-resistant nosocomial pathogens all over the world. Nonetheless, very little is known about the diversity of *A. baumannii* lineages coexisting in hospital settings. Here, using whole-genome sequencing, epidemiological data, and antimicrobial susceptibility tests, we uncover the transmission dynamics of extensive and multidrug-resistant *A. baumannii* in a tertiary hospital over a decade. Our core genome phylogeny of almost 300 genomes suggests that there were several introductions of lineages from international clone 2 into the hospital. The molecular dating analysis shows that these introductions happened in 2006, 2007, and 2013. Furthermore, using the accessory genome, we show that these lineages were extensively disseminated across many wards in the hospital. Our results demonstrate that accessory genome variation can be a very powerful tool for conducting genomic epidemiology. We anticipate future studies employing the accessory genome along with the core genome as a powerful phylogenomic strategy to track bacterial transmissions over very short microevolutionary scales.

**IMPORTANCE** Whole-genome sequencing for epidemiological investigations (genomic epidemiology) has been of paramount importance to understand the transmission dynamics of many bacterial (and nonbacterial) pathogens. Commonly, variation in the core genome, single nucleotide polymorphisms (SNPs), is employed to carry out genomic epidemiology. However, at very short periods of time, the core genome might not have accumulated enough variation (sufficient SNPs) to tell apart isolates. In this scenario, gene content variation in the accessory genome can be an option to conduct genomic epidemiology. Here, we used the accessory genome, as well as the core genome, to uncover the transmission dynamics of extensive and multidrug-resistant *A. baumannii* in a tertiary hospital for a decade. Our study shows that accessory genome variation can be a very powerful tool for conducting genomic epidemiology.

**KEYWORDS** genomic epidemiology, bacterial clones, transmission dynamics, accessory genome, *Acinetobacter baumannii*, molecular epidemiology

Antimicrobial drug resistance is one of the most important health issues worldwide. The ESKAPE (*Enterococcus faecium*, *Staphylococcus aureus*, *Klebsiella pneumoniae*, *Acinetobacter baumannii*, *Pseudomonas aeruginosa*, and *Enterobacter* species) group have many acquired antimicrobial resistance genes (ARGs) and are a major source of

Address correspondence to Santiago Castillo-Ramírez, iago@ccg.unam.mx.

Accessory genomic epidemiology of bacterial pathogens. Our study demonstrates that accessory genome variation can be a very powerful tool for conducting genomic epidemiology.

deadly infections all over the world (1). The World Health Organization list of bacterial pathogens urgently requiring novel antibiotics ranked *A. baumannii* at the highest priority status (priority 1: critical) (2). Importantly, many *A. baumannii* nosocomial infections are due to multidrug-resistant (MDR) isolates, defined as isolates resistant to at least one agent in three or more antimicrobial drug classes (3). Furthermore, extensively drug-resistant isolates (XDR), defined as resistant to at least one agent in all but one or two antimicrobial drug classes (3), have been also reported. Clearly, due to these MDR and XDR phenotypes, *A. baumannii* is a global public health issue that needs to be seriously considered.

Over the last decades, whole-genome sequencing (WGS) has become the ultimate approach to study the dissemination of many bacterial pathogens (4–8). WGS has been extremely useful in the case of *A. baumannii*, where the very dynamic nature of its genome renders the two multilocus sequence typing (MLST) schemes faulty (9, 10). Specifically, the Oxford MLST scheme has some sequence types (STs) that do not form monophyletic groups, whereas the Pasteur MLST scheme has low resolution (9, 10). WGS has been used to study the spread of clones of *A. baumannii* at a national level and even at the continental level (11, 12). Over the last years, WGS has been used to analyze clone diversity within and between hospital settings in several parts of the world (13–21). For instance, there have been studies using WGS to study clones of this species within hospitals in some countries, such as Brazil, Lebanon, Malaysia, United Kingdom, and Vietnam (13, 18, 19, 21). However, very little is known about different lineages causing MDR infections in single hospital settings in Mexico and Central America. Nonetheless, a recent study has suggested that different lineages could be cocirculating in the National Institute of Oncology in Mexico City, Mexico (11).

When using WGS for epidemiological and phylogeographical investigations, the most common approach is to use core regions or core single nucleotide polymorphism (SNP) phylogenies and SNP differences between isolates (13–21). Phylogenies based on either core SNPs or core regions have been effective in most cases. However, considering very short periods of time, sometimes hardly any SNPs have accrued between strains, at least in the core genome. For instance, in a recent study (22) that analyzed 24 *A. baumannii* strains collected from the blood of a single patient (over a period of 6 months), some strains (sampled during the same month) were identical considering their core SNPs. However, a few years ago, in a population genomics study, analyzing a set of very closely related isolates from the same place, we showed that acquisition and depletion of genes occur considerably faster than the accumulation of mutations in *A. baumannii* (23). Therefore, this gene content variability can be very valuable for studying the evolutionary relationships of isolates at very short periods of time.

The goal of our study was to use WGS, along with epidemiological data and antimicrobial susceptibility testing, to analyze the diversity of clones (and their antimicrobial resistance patterns) in a hospital sampled for a decade. To that end, we carried out a genomic epidemiology study to characterize the lineages of *A. baumannii* in a tertiary, teaching hospital in Guadalajara (Hospital Civil de Guadalajara [HCG]), Mexico. Notably, besides the core genome, we also used the accessory genome to establish the dissemination of lineages across different wards.

## RESULTS

**Extensive and multidrug drug-resistant lineages from international clone 2.** We sequenced 73 isolates of *A. baumannii* sampled between 2007 and 2017 from the HCG; these isolates were identified in previous studies (24, 25), and all belonged to the same pulse type, which was the most prevalent clone in the hospital (see Table S1 in the supplemental material). We used the genome sequences to conduct sequence type (ST) assignation, considering both the Oxford and Pasteur MLST schemes (26, 27). These isolates belonged to just four STs (Fig. 1) as per the Oxford scheme. ST417 was the most frequent with 38 isolates, followed by ST208 having 23 isolates; whereas ST136 and ST369 had 7 and 5 isolates, respectively (Table 1). Of note, all these STs are

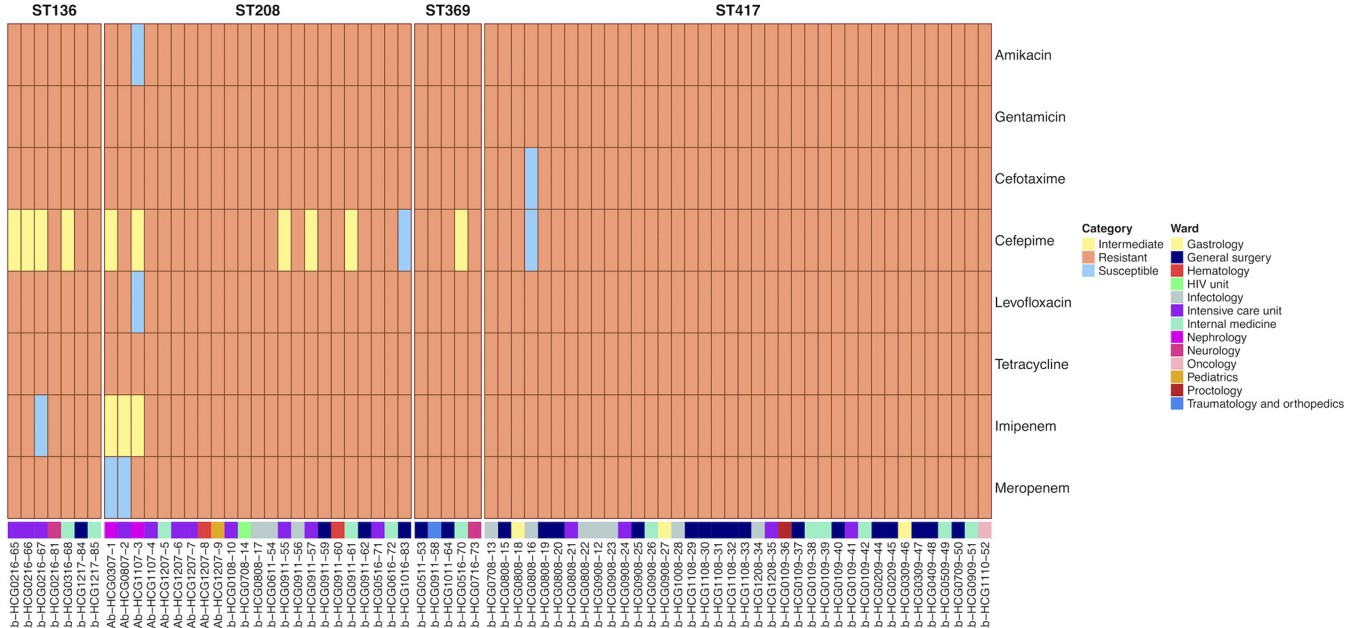

**FIG 1** Antibiotic resistance patterns of the 73 *A. baumannii* isolates from the HCG. Different drug classes were tested, and the actual MICs for each isolate and each drug are provided in Table S2 in the supplemental material. For each isolate, we also provide the ward from which it was sampled. The ST assignation for the isolates is at the top in bold. Re the strain names, the numbers after the "HCG" initials give the month (first two digits) and the year (third and fourth digits) of sampling. For instance, Ab-HCG0216-65 was sampled in February 2016.

part of the international clone 2 (IC2) (24, 28–30). We conducted antimicrobial susceptibility tests on the isolates for eight antibiotics: amikacin, cefepime, cefotaxime, gentamicin, imipenem, levofloxacin, meropenem, and tetracycline.

Table S2 gives the minimum inhibitory concentration (MIC) values for each isolate and each drug tested. Remarkably, all the isolates were resistant to many antibiotics (Fig. 1 and Table 1). All the isolates were resistant to gentamicin and tetracycline. Considering meropenem, all the isolates from ST136, ST369, and ST417 were 100% resistant, and 91.3% of the ST208 isolates were resistant. Regarding imipenem, all the isolates from ST369 and ST417 were 100% resistant, but isolates from ST208 and ST136 were 87% and 85.7% resistant, respectively. ST369 and ST417 had more resistance than ST136 and ST208 (Table 1 and Fig. 1). All but one of the ST369 isolates were resistant to all the antibiotics tested; whereas for the ST417 isolates also, all but one were resistant to all the antibiotics tested (Fig. 1). Most of the isolates, irrespective of their ST assignation, were XDR. There were a few isolates that were MDR isolates within ST208. These results suggest that these isolates, regardless of which ST they belong to, are either MDR or XDR. Taken together, these results show that different MDR and XDR lineages (STs), all assigned to IC2, were found in the hospital.

**Several introductions into the hospital between 2004 and 2015.** To have a better idea of the introduction of these STs into the hospital, we downloaded high-quality

**TABLE 1** Summary of the isolates collected from the HCG

| Ox. ST[a] | No. of isolates | No. of wards[b] | Isolation date | % of resistant isolates resistant against the following antibiotic[c]: | | | | | |
|---|---|---|---|---|---|---|---|---|---|
| | | | | AMK | CTX | CEP | LVX | IPM | MEM |
| 136 | 7 | 4 | 2016, 2017 | 100 | 100 | 42.9 | 100 | 85.7 | 100 |
| 208 | 23 | 8 | 2007, 2008, 2011, 2016 | 95.7 | 100 | 73.9 | 95.7 | 87 | 91.3 |
| 369 | 5 | 4 | 2011, 2016 | 100 | 100 | 80 | 100 | 100 | 100 |
| 417 | 38 | 7 | 2008, 2009, 2010 | 100 | 97.4 | 97.4 | 100 | 100 | 100 |

[a]Ox. ST, sequence type by the Oxford MLST scheme.
[b]Number of different wards in which the isolates were sampled.
[c]Percentage of isolates resistant against different antibiotics. The antibiotics tested were amikacin (AMK), cefotaxime (CTX), cefepime (CEP), levofloxacin (LVX), imipenem (IPM), meropenem (MEM), gentamicin, and tetracycline. All the isolates were 100% resistant against gentamicin and tetracycline.

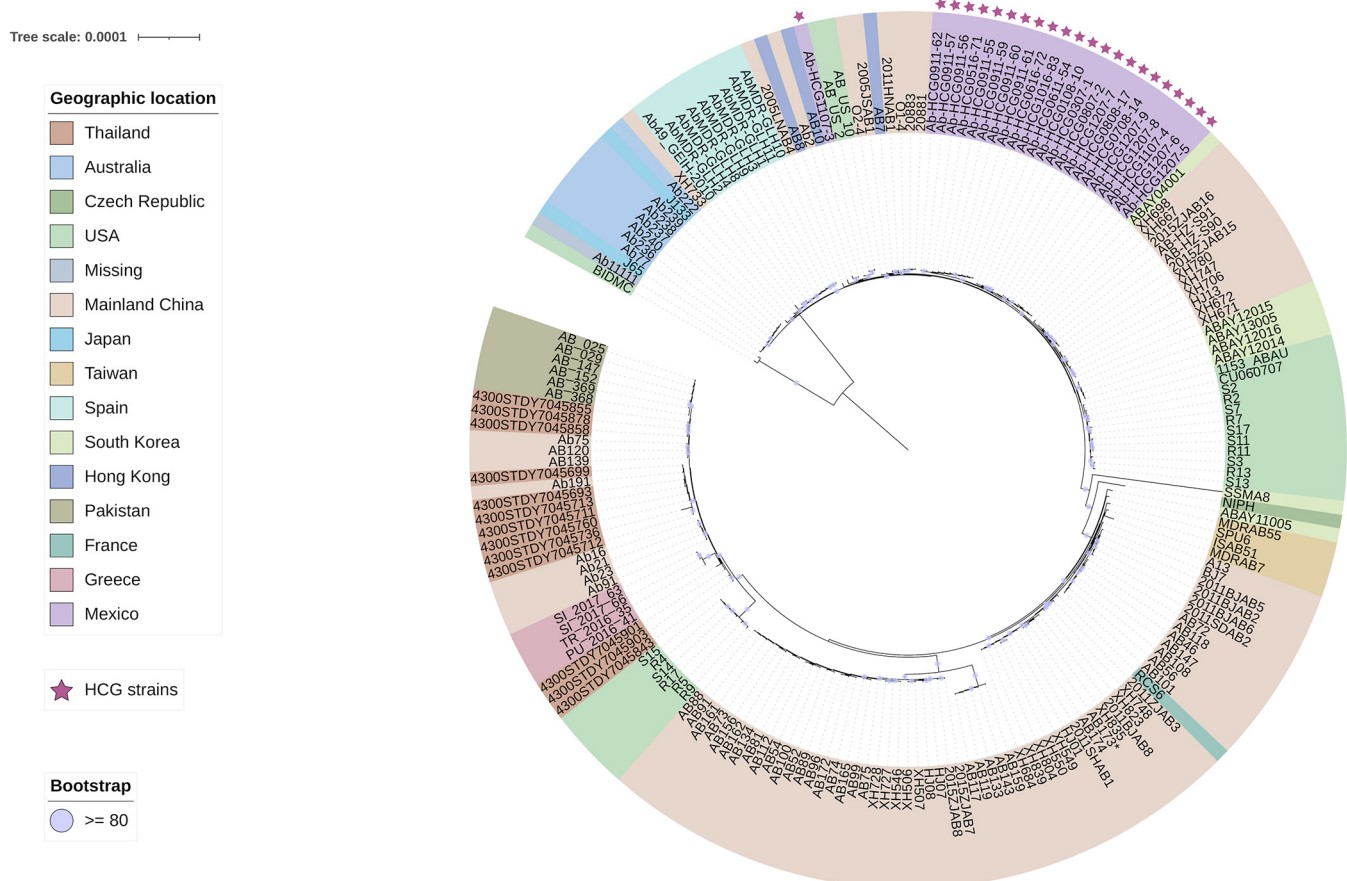

**FIG 2** Core genome phylogeny for ST208. The phylogeny is annotated with the geographic region. The stars mark the isolates from the HCG. Bootstrap values equal to or higher than 80 are shown with violet dots at the internal nodes of the tree. The bar gives the number of substitutions per site.

publicly available genomes that were assigned to the same STs as those we found in the HCG isolates. Then, considering the HCG isolates and the publicly available genomes, we constructed a maximum likelihood core genome phylogeny (see Materials and Methods). This phylogeny is shown in Fig. S2 in the supplemental material. The phylogeny shows that the STs were introduced separately into the hospital (see stars in Fig. S2). The HCG isolates of the most frequent ST, namely, ST417, all clustered together. Given the monophyletic nature of this ST in the hospital, a single introduction of this ST into the hospital is the most plausible explanation.

We also conducted core genome phylogenies individually for ST136, ST208, and ST369. In the case of HCG ST208 isolates, there were two introduction events (Fig. 2). We noted that all the isolates but isolate Ab-HCG1107-3 cluster together, implying one single introduction event, whereas Ab-HCG1107-3, located on a distant branch, was introduced independently. Although the core genome phylogeny for the ST136 isolates (Fig. 3, top panel) appears to suggest the monophyly of HCG isolates, the global phylogeny that includes all the STs (Fig. S2) clearly shows that the HCG ST136 isolates form a paraphyletic group as two isolates (Ab-HCG0516-70 and Ab-HCG0716-73) assigned to the ST369 clustered with the HCG ST136 isolates. However, all the HCG ST136 isolates plus the two ST369 isolates mentioned above were introduced in one single event into the hospital (Fig. S2). Of note, also the core genome phylogeny for the ST136 isolates is misleading, as it shows that the ST136 isolates from the HCG form a monophyletic group; however, the global phylogeny clearly demonstrates that this is not the case.

Then, we wanted to know the time of introduction of these STs into the hospital. Thus, we conducted a molecular dating analysis (see Materials and Methods). We only

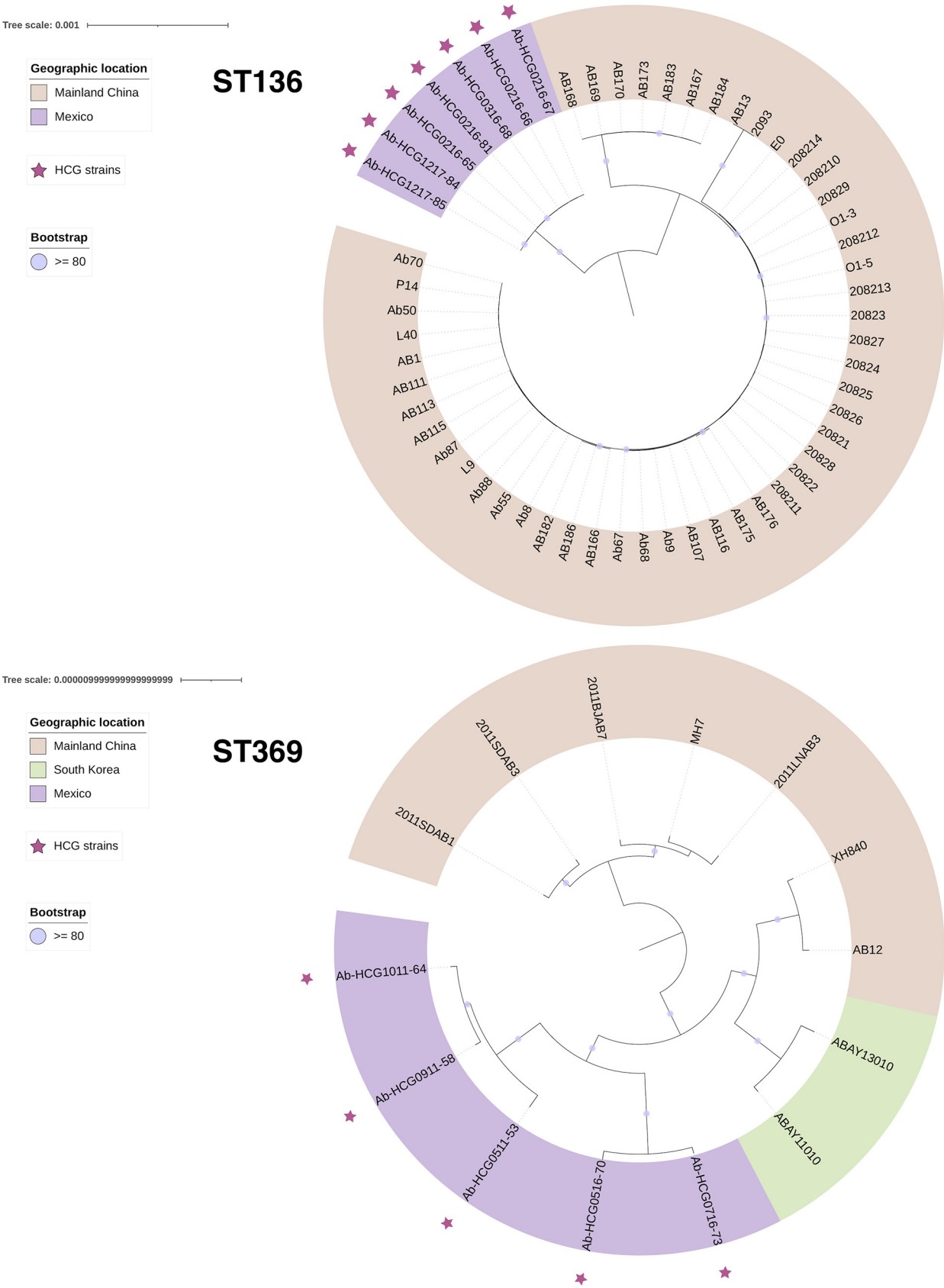

**FIG 3** Core genome phylogenies for ST136 and ST369. The phylogenies are annotated with the geographic region. The stars mark the isolates from the HCG. Bootstrap values equal to or higher than 80 are shown with violet dots at the internal nodes of the tree. The bars give the number of substitutions per site.

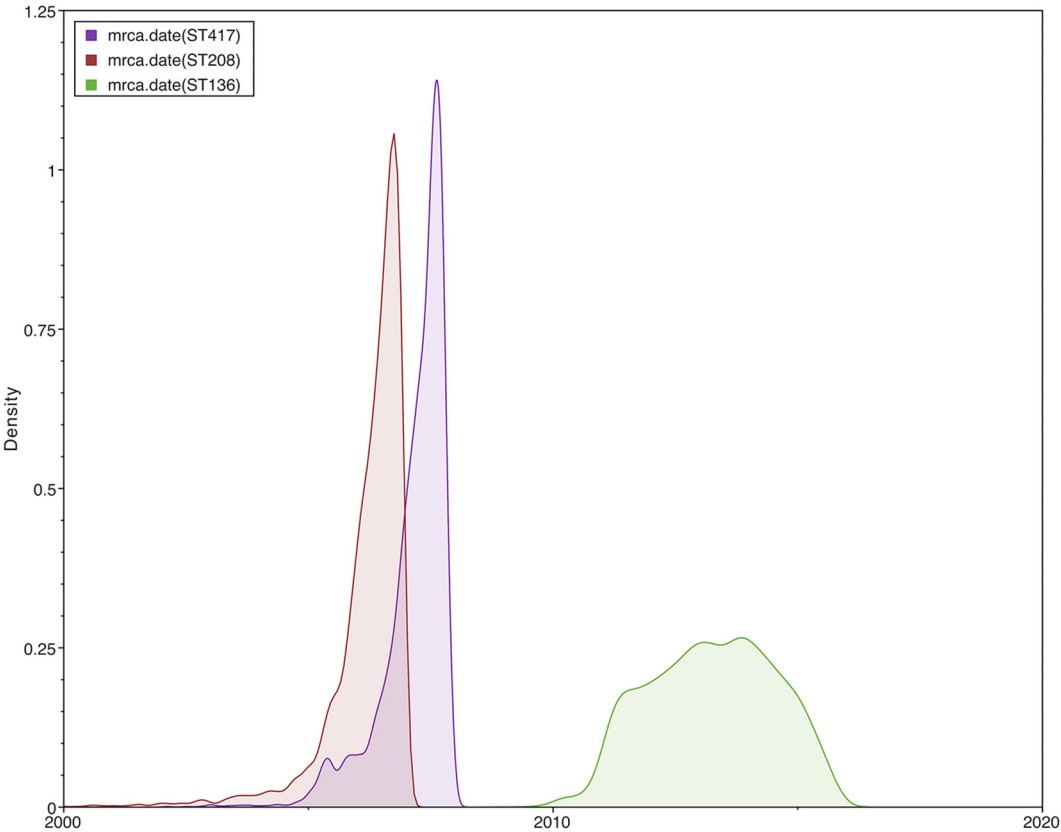

**FIG 4** Molecular dating analysis of the main STs. Marginal posterior density for the tMRCA for ST136 (green), ST208 (red), and ST417 (purple). The *x* axis shows the time between 2000 and 2020.

estimated the time to the most recent common ancestor (tMRCA) for ST417, ST208 (not including the isolate introduced in the independent event), and ST136. Figure 4 shows the marginal posterior densities of the tMRCAs for the three STs considered. From this figure, it is clear that ST208 and ST417 were introduced very close to one another, whereas ST136 was introduced a few years later. ST208 was the first lineage to reach the hospital, and it was introduced in early 2006, 95% high posterior density interval (95% HPD), mid-2004 to late 2006. ST417 was introduced about 1 year later, in early 2007 (95% HPD, late 2005 to late 2007). Some 5 years later, ST136 was introduced (tMRCA, 2013; 95% HPD, early 2011 to mid-2015). Taking into account the 95% HPDs for the three STs, these lineages were introduced into the hospital between 2004 and 2015. Collectively, these data suggest these STs were introduced into the HCG in five events and that these introductions occurred more or less over a decade.

**High transmission across wards.** The isolates sequenced were collected from 13 different wards in the hospital, and all the STs were found in several wards (Table 1). Figure 1 shows the distribution of isolates per ward and year. Intensive care unit (ICU) and General Surgery were the most represented wards among the isolates sequenced. To further analyze the transmission dynamics, we conducted two analyses just comparing isolates from the same ST. In each analysis, we compared isolates between and within wards and counted the number of differences. In the first analysis, we counted the number of core SNPs telling apart isolates, whereas in the second analysis, we counted the number of genes differing between isolates (Fig. 5). These analyses were conducted only for ST417 and ST208, the two most frequent STs. If isolates were mainly disseminating in the same ward, one would expect considerably fewer differences (either core SNPs or number of genes) when comparing isolates from within the same ward than when comparing isolates from different wards. In contrast, if there

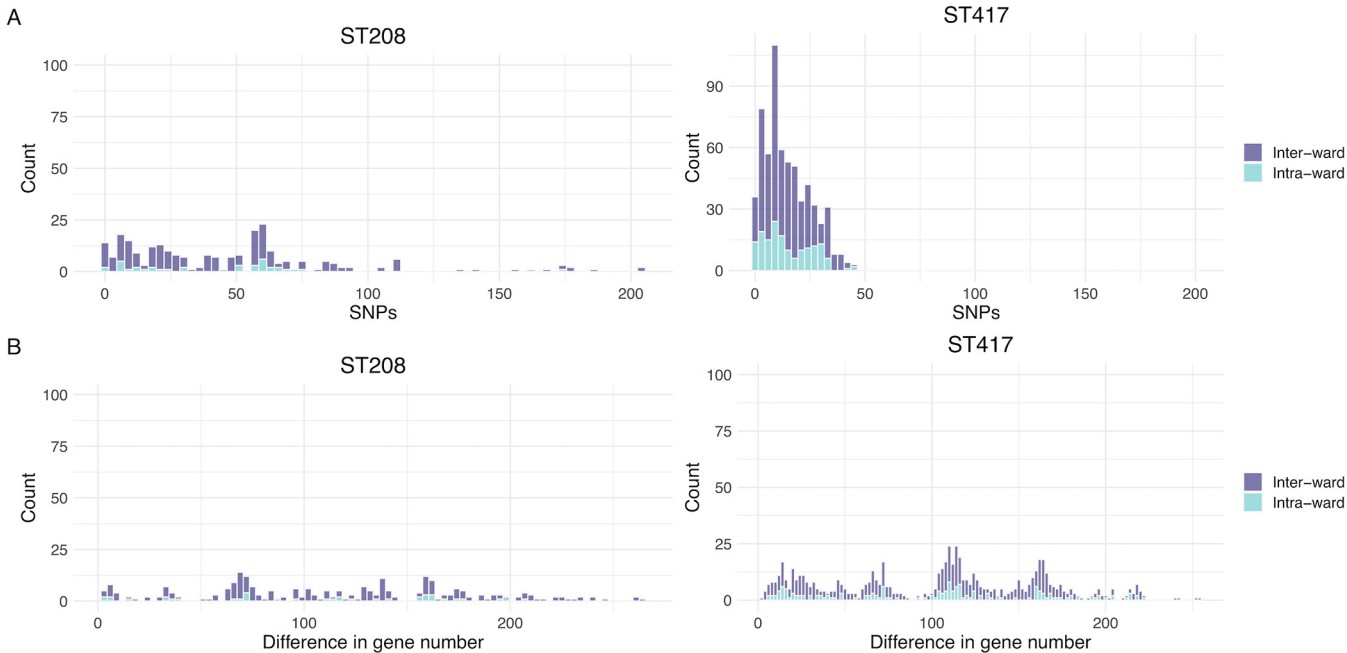

**FIG 5** Between- and within-ward transmission analysis. We conducted pairwise comparisons for the isolates of ST208 (left-hand side) and ST417 (right-hand side), which were the most abundant STs. Comparisons considering isolates from the same ward are shown in cyan (light blue) bars, whereas comparison of isolates from different wards are the purple bars. The top panels give the differences in terms of core SNPs, whereas the bottom panels show the differences in gene content.

were high transmission across wards, the number of differences when comparing within-ward versus between-ward isolates would not be different. Our analyses supported the latter scenario, whether we considered core SNPs (top panels) or the number of genes differing between isolates (bottom panels). Considering both measures, the distribution of within-ward comparisons overlaps with the distribution of between-ward distribution (Fig. 5). We did not find significant differences comparing the distributions of core SNPs (Wilcoxon-Mann-Whitney test, ST417 $P$ value = 0.6872 and ST208 $P$ value = 0.7834) or comparing the distribution of different genes (Wilcoxon-Mann-Whitney test, ST417 $P$ value = 0.6568 and ST208 $P$ value = 0.6478). To further explore the high transmission of isolates across wards and to compare the discriminative typing power of the core genome and the accessory genome, we constructed another two trees just including the 73 HCG isolates. One tree was constructed using the core genome, and the second tree was based on a gene content distance matrix (see Materials and Methods). These two trees are presented in Fig. 6. Both trees reinforce the idea of high transmission across wards, as isolates for any given ward very often had their closest related isolate coming from a different ward. Of note, the core genome tree (Fig. 6B) had several polytomies, and these affect the different STs, ST417 and ST208 being the most outstanding cases. Thus, some isolates were identical to other isolates under the core genome tree. In contrast, the phylogenetic tree based on gene content (Fig. 6A) showed far greater resolution. In this tree, all the isolates could be differentiated from one another. Therefore, gene content variation (accessory genome) had more discriminative genotyping resolution than the core genome. All in all, these results demonstrate that there is frequent transmission of isolates across many different wards in the hospital. Furthermore, the analysis of the accessory genome was of paramount importance to describe the transmission dynamics of these isolates.

## DISCUSSION

As with many other bacterial pathogens, WGS has been instrumental in properly understanding the transmission and phylogeography of many lineages of *A. baumannii* in several countries (13–21). WGS has also been employed to fully appreciate the

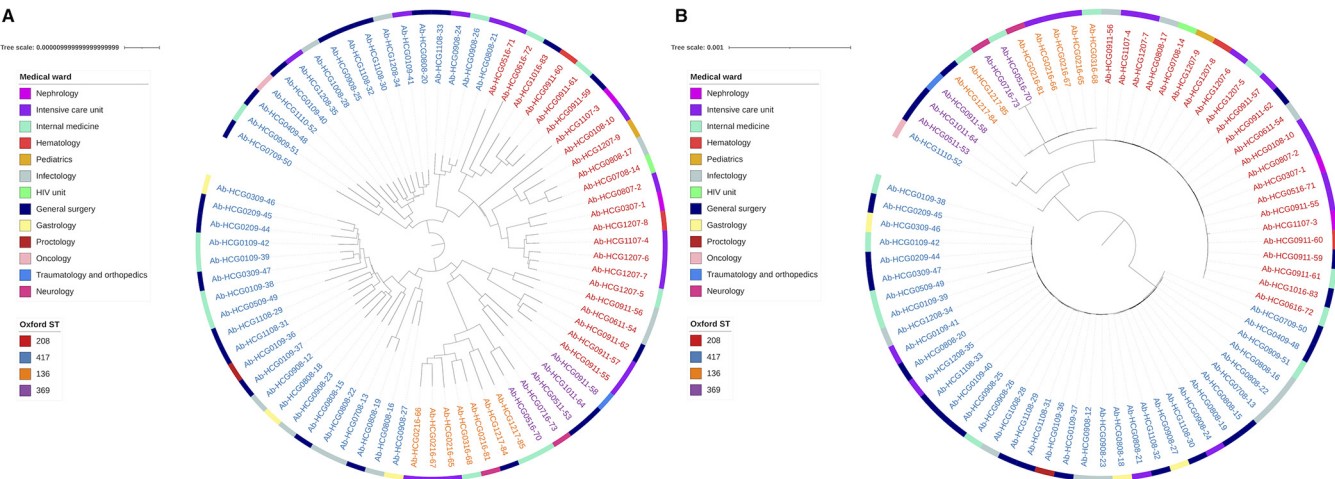

**FIG 6** Accessory and core genome trees for the HGC isolates. Both trees were constructed considering only the newly sequenced 73 *A. baumannii* isolates from the HCG. (A) Neighbor-joining tree based on a gene content distance matrix. (B) Maximum likelihood (ML) tree based on the alignment used for the ML phylogeny in Fig. S2. The alignment was edited to contain only the 73 isolates from the HCG. The isolates are color coded by ST, whereas the external ring provides the wards for the isolates.

processes generating genomic variation within the species (23, 31, 32). Even a special issue has been published recently about the impact of genomics on better comprehending antibiotic resistance and virulence, not only in *A. baumannii* but also the whole genus (33).

In this study, we used both core and accessory genome to get an uber-resolution of the transmission dynamics of cocirculating lineages in a tertiary hospital over a decade. We discovered four lineages (ST136, ST208, ST369, and ST417), and all were from the IC2 (24, 28–30). This clone has been extensively studied and has been found in many parts of the world. However, this clone has not been described as one of the most relevant clones in Mexico in recent genomic studies (11, 23). Nonetheless, IC2 has been described in other countries in Latin America (34) or the United States (35), which shares an extensive border with Mexico. Our phylogenetic analysis based on the core genome indicates that these STs were introduced in different events into the HCG. These introduction events occurred in 2006 and 2007 and in 2013. Clinically speaking, it is relevant that the isolates from these STs were either MDR or even XDR. Also, most of these isolates were resistant to carbapenems. In contrast to the idea of just having one predominant clone at any time, we found different STs in the period of time this study encompasses. Hence, it seems that different lineages can be coexisting within the same hospital setting.

The distribution of isolates per ward suggests that the different STs have spread to many wards in the hospital. To properly analyze dissemination in the hospital wards, we used both core and accessory genome. Although the accessory genome analysis had considerably more resolution, both analyses clearly showed that the STs have been extensively disseminated in many wards. Thus, there is no population structuring according to wards. From a practical point of view, these results suggest that infection control measures were not able to contain the dissemination of these STs in the hospital. In accordance with our previous work (9), we noted that several STs (ST369 and ST136) have problems (i.e., they do not form monophyletic groups) properly genotyping the isolates assigned to them. This highlights the need for the use of more powerful genotyping approaches for *A. baumannii*. Clearly, if possible, WGS should be the method of choice. Over the last decade, most bacterial genomic epidemiology studies have used core SNPs to reconstruct the transmission dynamics of pathogens. These studies have assumed that *de novo* mutations accrue faster than the transmission events inferred. However, at very short time scales, we cannot judge whether mutations or transmission events occur faster. To overcome this, we used gene content variation, which occurs faster than mutations (23), to have a higher genotyping resolution

at very short time scales. Our results demonstrate that this type of alternative genomic variation can be used to track the transmission events over very short periods of time. Ideally, one can use both core and accessory genome to conduct genomic epidemiology at an unprecedented level, thus going beyond what the core genome variation can resolve by itself. We think the use of the accessory genome (as an extra phylogenomic marker) can be particularly useful for epidemiological investigations of outbreaks.

In conclusion, our study shows that both the accessory genome and the core genome can be very powerful tools to understand the transmission dynamics of bacterial pathogens. We anticipate our study to be a reference point for more elaborate analyses using the accessory genome along with the core genome as a very useful phylogenomic approach to investigate bacterial transmission dynamics over very short microevolutionary scales.

## MATERIALS AND METHODS

**Study design, settings, and isolates.** The Hospital Civil de Guadalajara (HCG) is a tertiary, teaching hospital in Guadalajara, Jalisco (Western Mexico), and it has 899 beds. The initial collection had 130 clinical isolates of *Acinetobacter baumannii*, which were identified in previous studies (24, 25), and these isolates covered a decade, from 2007 to 2017. All the isolates came from positive cultures of clinical samples taken from hospitalized patients. In all the cases, *A. baumannii* was the etiological agent of the infection, and there was no evidence that indicated colonization or community-acquired infection in any patient. Only one isolate was taken from each patient. All but three isolates were pulse type 22. We sequenced 76 isolates (73 pulse type 22 isolates and 3 isolates assigned to ST758 as per Oxford MLST) trying to represent as much as possible the proportion of isolates per year in the initial collection. We included only the 73 pulse type 22 isolates for downstream analyses; the details (date, ST, ward, etc.) of these genomes are shown in Table S1 in the supplemental material.

**DNA extraction and genome sequencing.** Genomic DNA was extracted using the QIAamp DNA mini-kit (Qiagen, Hilden, Germany) according to the manufacturer's instructions from a single colony grown in LB medium grown overnight at 37°C (36). The genome sequencing of the isolates was carried out at the Instituto Nacional de Medicina Genómica (https://www.inmegen.gob.mx/) in Mexico City, Mexico. A $2 \times 250$-bp configuration, employing an Illumina Miseq platform, was used. We trimmed the first and the last five bases in each read and the poor-quality bases with Trim Galore v0.6.4 (https://github.com/FelixKrueger/TrimGalore). We assembled the genomes with SPAdes v3.13.1 (37) (see Table S1 for assembly statistics), and then we annotated and checked the contamination and completeness of each genome as we did previously (38); we used Prokka v1.13 (39) for the annotation and CheckM v1.0.11 (40) for contamination and completeness. We used the PubMLST database (41) to assign sequence types, under both the Oxford and Pasteur MLST schemes (26, 27), to the newly sequenced isolates. We also downloaded over 200 genomes from the NCBI that had the same STs as those assigned to the newly sequenced isolates (see Table S1). All these publicly available genomes were of high quality, no genomes under the assembly level "contigs" were included.

**Antibiotic susceptibility testing.** The MICs for different drugs were performed for the 73 isolates sequenced using serial dilution on agar following the guidelines of the Clinical and Laboratory Standards Institute (CLSI) (42).

The antimicrobial agents tested against the 73 isolates include amikacin (AMK), gentamicin (GEN), cefotaxime (CTX), cefepime (FEP), levofloxacin (LVX), tetracycline (TET), imipenem (IPM), and meropenem (MEM). The MIC was determined as the lowest concentration of antibiotics in which the *A. baumannii* growth was inhibited. Table S2 gives the MIC values for each isolate and each drug tested. The classification of each isolate as susceptible or resistant was established according to the MIC breakpoints in the guidelines of the CLSI (42). *Escherichia coli* strain ATCC 25922 was employed for quality control tests.

**Core genome phylogenies.** We constructed a maximum likelihood core phylogeny including all the STs as in a previous study (11). Single gene families were identified analyzing the output of Roary v3.12.0 (43). We employed PhiPack (44) to identify recombination in each one of the single gene families. Single gene families without recombination (SGFwR) were concatenated and a phylogenetic tree was constructed using RAxML v8.2.12 (45). We annotated the phylogenetic tree with iTOL v6.2 (46). The same strategy was applied to ST136, ST208, and ST369 separately to construct a core phylogenetic tree for each of these STs. Finally, we also ran another tree with the alignment containing all the STs but only considering the 73 isolates from the HCG.

**Molecular dating analysis.** We implemented a molecular dating analysis using BEAST v2.6.2 (47). The analysis was run on the concatenated alignment of the SGFwR considering the 73 HCG isolates. We used TempEst v1.5.3 (48) to check the temporal signal within the data, and we excluded some of the isolates whose genetic divergence and sampling date were not congruent. The excluded isolates were Ab-HCG0509-49, Ab-HCG0516-70, Ab-HCG0516-71, Ab-HCG0616-72, Ab-HCG0716-73, Ab-HCG0808-18, Ab-HCG0911-56, Ab-HCG1016-83, Ab-HCG1107-3, and Ab-HCG1110-52. The list of isolates used for the analysis is provided in Text S1 in the supplemental material. We found a strong temporal signal, and the regression of dates of sampling against the root-to-tip distances was 0.9074 (see Fig. S1 in the supplemental material). We used the TrN+G as the site model, for this was selected by jModelTest v2.1.4 (49) as the best model under the Akaike information criterion. We used a log-normal relaxed molecular clock

model, which was calibrated using the sampling dates of the isolates. The analysis was run for 900,000,000 generations, and samples were taken every 900,000 generations. The analysis was run twice to make sure the results were consistent. In both analyses, we ensured that the effective sampling size of the likelihood of the tree was well above 200.

**Comparisons within wards versus across wards and gene content tree.** To evaluate the dissemination across wards, we employed two measures. One measure was the number of core SNPs between pairs of isolates. For this, we made use of all the SNPs found in the concatenated alignment of 2,488 SGFwR for only the 73 HCG isolates (we ran another pangenome analysis using Roary v3.12.0 just with the HCG isolates). For this analysis, we discarded the isolates Ab-HCG0509-49 and Ab-HCG0611-54 which seem to be hypermutator strains. The second measure counted the number of differing genes between pairs of isolates. For both measures, we just compared the isolates from within the same ST, and the pairwise comparisons were divided into within-ward and between-ward comparisons. We also constructed a gene content correlation matrix from the pangenome analysis of the 73 HCG isolates. Then, to create a neighbor-joining tree based on gene content, we created a gene content distance matrix (based on the gene content matrix) employing the *dist* function in R and setting Euclidean distance. After that, the APE v5.4.1 library (50) was used to construct a neighbor-joining tree on the distance matrix.

## SUPPLEMENTAL MATERIAL

Supplemental material is available online only.

**TEXT S1**, TXT file, 0 MB.
**FIG S1**, PDF file, 0.1 MB.
**FIG S2**, PDF file, 0.2 MB.
**TABLE S1**, XLSX file, 0.04 MB.
**TABLE S2**, XLSX file, 0.01 MB.

## ACKNOWLEDGMENTS

We are grateful to Victor Manuel del Moral Chávez and Alfredo José Hernández Álvarez for their valuable job installing some of the bioinformatic programs used in this study. We also thank Joel Gómez Espíndola, Iván Uhthoff Aguilera, Maria Gabriela Guerrero Ruiz, and Luis Fernando Lozano Aguirre Beltrán for technical support on several matters. We are also thankful to Catalina Gayosso Vázquez for her technical assistance on the antibiotic susceptibility assays.

This work was financed by CONACyT Ciencia Básica 2016 (grant 284276) and "Programa de Apoyo a Proyectos de Investigación e Innovación Tecnológica PAPIIT" (grant IN206019); these two grants were awarded to S.C.-R. V.M.-E. is a Ph.D. student from the Programa de Doctorado en Ciencias Biomédicas, Universidad Nacional Autónoma de México (UNAM), and she holds a CONACYT doctoral fellowship (1005234).

S.C.-R. conceived and supervised the study. V.M.-E. conducted almost all the *in silico* analyses. S.C.-R. conducted the molecular dating analysis. M.D.A.-C. supervised the DNA extraction and antibiotic susceptibility testing. J.L.F.-V. carried out DNA extraction. J.M.-M. conducted antibiotic susceptibility testing. M.D.A.-C., E.R.-N., and R.M.-O. provided the isolates and metadata. I.L.H.-G. downloaded the publicly available genomes and assigned the STs for the genomes.

We have no potential conflicts of interest.

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
