## [Reviewer comments · mSystems]

Accessory genomic epidemiology of co-circulating *Acinetobacter baumannii* clones

Valeria Mateo-Estrada, Jose Fernandez-Vazquez, Julia Moreno-Manjón, Ismael Hernández-González, Eduardo Rodríguez-Noriega, Rayo Morfín-Otero, Maria Alcántar-Curiel, and Santiago Castillo-Ramírez

Corresponding Author(s): Santiago Castillo-Ramírez, Programa de Genómica Evolutiva, Centro de Ciencias Genómicas, Universidad Nacional Autónoma de México

Review Timeline:

Submission Date:	May 20, 2021
Editorial Decision:	June 21, 2021
Revision Received:	June 28, 2021
Accepted:	June 28, 2021

Editor: Jack Gilbert

Reviewer(s): Disclosure of reviewer identity is with reference to reviewer comments included in decision letter(s). The following individuals involved in review of your submission have agreed to reveal their identity: Xiaoting Hua (Reviewer #3)

Transaction Report:

DOI: <https://doi.org/10.1128/mSystems.00626-21>

Dr. Santiago Castillo-Ramírez
Programa de Genómica Evolutiva, Centro de Ciencias Genómicas, Universidad Nacional Autónoma de México
Evolutionary Genomics Research Program
CCG UNAM
Cuernavaca 62210
Mexico

Re: mSystems00626-21 (Accessory genomic epidemiology of co-circulating *Acinetobacter baumannii* clones)

Dear Dr. Santiago Castillo-Ramírez:

I have received the reviews of your manuscript entitled "Accessory genomic epidemiology of co-circulating *Acinetobacter baumannii* clones", and I regret to inform you that we will not be able to publish it in mSystems. Your submission was read by reviewers with expertise in the area addressed in your study. One of them raised important concerns that question both methodology and main conclusions of the manuscript. I personally share the reviewer's view that your paper does not meet the standards necessary for publication. Copies of the reviewers' comments are attached for your consideration.

While this manuscript may not be a good fit for mSystems, I feel that the work has merit and would like to offer you the opportunity to transfer this manuscript to Microbiology Spectrum. Microbiology Spectrum is a new open-access journal from the ASM that seeks to publish technically sound, primary research across the entire range of microbial sciences and allied fields. More information on Microbiology Spectrum can be found here (<https://www.asmscience.org/content/journal/microbiolspec>). The Academic Editors at Microbiology Spectrum welcome format-neutral submissions and offer fast-turnaround from submission to peer review and final online publication. As your article was reviewed at mSystems, you can transfer the paper along with the reviews and reviewer identities to Microbiology Spectrum. The Editors at Microbiology Spectrum often offer expedited acceptance without further peer review.

Please use the link below to transfer your paper to Microbiology Spectrum. Once you transfer the paper, Spectrum staff will contact you and you will be given the opportunity to update files and upload responses to reviewers etc. If you would like any further information or input prior to transfer, please write to Anand Balasubramani, Managing Scientific Editor (spectrum@asmusa.org).

Please note that the transfer link below will be visible only in the decision letter sent to the corresponding author.

If you would like to transfer the manuscript and the mSystems reviews to Spectrum, please use this link:

Link Not Available

I regret that I cannot accept your manuscript for publication. Please be assured that the journal would welcome your future submissions.

Thank you for considering mSystems.

Sincerely,

Pedro Oliveira
Editor, mSystems

Reviewer comments:

Reviewer #3 (Comments for the Author):

all questions were addressed.

Reviewer #4 (Comments for the Author):

Mateo-Estrada et al. present a genomic analysis of 73 *Acinetobacter baumannii* isolates collected between 2007-2017 from a tertiary hospital in Mexico. Their main goal was to assess diversity in the hospital over this time and evaluate antimicrobial susceptibility and introduction/transmission of the predominant lineages. One of the most promising aspects of this paper was the use of the accessory genome, which is often overlooked.

Unfortunately, I do not think this study was able to achieve all that it promised in the abstract. There was a general lack of specificity and detail throughout the manuscript, which made it difficult to completely assess the work. I was not convinced by the methods, again potentially due to a lack of information but also because there seemed to be no attempt to validate the results or investigate unexpected outcomes. Conclusions seem to be drawn with very little support or explanation. Generally, the scope of literature referred to in this study appears narrow to me and ignores a large body of work contributed globally by other researchers looking at WGS and *A. baumannii*. I also found the text repetitive and vague in parts and should be made more concise.

Introduction:

Line 72: should reference the definition of MDR and XDR. Sentence structure would also sound better if said "...due to multidrug resistant (MDR) isolates, defined as resistance to..."

Line 82: I would suggest the authors look into this publication and cite this instead, as it provides a much more in-depth analysis into the issues surrounding the *A. baumannii* MLST schemes:
<https://doi.org/10.3389/fmicb.2019.00930>

Line 84: "WGS has been used to study the spread of clones of *A. baumannii* at national and even at continental levels (9, 10). However, unlike other important bacterial pathogens, WGS has hardly been used to analyse clone diversity within hospital settings. There have been some studies using WGS to study clones of this species within hospitals in some countries, such as UK, Lebanon and Vietnam (11-13). However, very little is known about different lineages causing MDR infections in single hospital settings in Latin America."

I do not agree with some of the statements made here. There are many studies that have made use of WGS to study the global phylogeny of *A. baumannii* and its diversity/transmission within hospital settings. The below is the result of a fairly brief search on the matter:

<https://doi.org/10.1128/AAC.00840-20>
<https://doi.org/10.1099/mgen.0.000052>
<https://doi.org/10.1016/j.ijid.2020.09.123>
<https://doi.org/10.1128/JCM.01818-15>
<https://doi.org/10.1128/mSphere.00934-19>
<https://doi.org/10.1128/AAC.02014-15>
<https://doi.org/10.1073/pnas.1104404108>
<https://doi.org/10.1099/mgen.0.000530>

As such, I think it is untrue to say that WGS has "hardly been used". Even the last statement about knowledge of circulating lineages in Latin America seems somewhat untrue, based on this study from Brazil (<https://doi.org/10.1016/j.ijantimicag.2020.106195>). I would encourage the authors to explore the existing literature more thoroughly in order to reconstruct their introduction to more accurately represent the context of this study.

Lines 94-97: please provide references

Materials and Methods:

Line 118: The hospital information and isolate information should probably be in a separate section called "study design, settings and isolates". In fact, given that this paragraph is quite long, I would recommend splitting this into separate sections for each (DNA extraction, genome sequencing and susceptibility testing).

Line 120: You refer to table 1 here, but this only includes the final 73 isolates, so I think is inappropriate in this sentence.

Line 122: How many of the original 134 were pulse type 22? Why were these 76 chosen? It is not clear if there was something more systematic than "trying to represent...the proportion of isolates per year...". Were there any conditions on not having repeat isolates from patients? Having a spread of samples throughout the year? Is there a reason for the large gap in samples from 2012-2015? It may be useful to include in supplementary table 1 the other isolates that were not selected for WGS.

Line 124: I would mention the three isolates being removed in a separate section where you describe how you performed quality control. This would also allow you to mention what exactly it was about the samples that resulted in their exclusion

Line 131: Please add version numbers for ALL software (this comes up throughout). What qualifies as a "poor-quality" base?

Line 135: In my opinion it is insufficient to just refer to another paper for methods. I think this should be followed up by a brief description of at least the software and versions used. This also allows the creators of the software to be properly credited, rather than re-citing another paper.

Line 136: Both the Oxford and Pasteur MLST schemes have references, which you can find on the

pubMLST website.

Line 155: Given that recombination analysis is one line, it can just be put in this section.

Line 169: "we excluded some of the isolates whose genetic divergence and sampling date were unusual."

Which isolates were excluded and why they were "unusual" needs to be described better here, otherwise it just seems like you were fitting your data to suit the model.

Line 184: Were you looking at core single gene families? Do you have an estimate of the genome size/percent from which you were calling SNPs?

Further to this, I am wondering if your SNP resolution was limited based on this method, and why you chose this method over SNP calling using a single whole genome reference for each ST and read mapping (with Snippy, for example). This would allow for more robust evaluation of each SNP site given the mapping quality and pileup, and would not be affected by possible artefacts introduced with assembly and gene annotation.

Results:

Line 210: specify which scheme you mean when saying "belonging to just four STs"

Line 231: Did you look at the genomic mechanisms for resistance (genes or SNPs)?

Line 238-256:

I have a few comments for this section:

"The phylogeny clearly shows that the STs were introduced in independent events into the hospital."

I would remove this sentence. It's not clear what you mean by "independent events" - it seems like you mean each ST was introduced once and proliferated from there, but you actually go on to describe isolates of the same ST being introduced separately. This sentence also relies heavily on the reader being able to see the phylogenetic tree, which is hard to do since it is in the supplementary.

Single introduction of ST417 and most of ST208: I am interested to know what further evidence you have that this was indeed a single introduction, and not a lineage in the community or separate healthcare setting causing repeat introductions into the hospital. Were all patients from which these isolates were taken negative on arrival (i.e. all samples were true hospital acquisitions?). Any environmental evidence? This sort of detail may need to be added to the methods section describing the isolates.

I think you need to spend more time analysing your ST369 and ST136 isolates. It is quite odd in your large tree that you have some ST369 clustering at opposite ends of the tree. Can you explain why? How different is the MLST scheme for ST369 to ST136? The Oxford scheme is notorious for including a capsule gene, which results in different ST assignments due to capsule switches.

I would say generally that given you have limited geographical context from other South American isolates, support for your "single introduction" events here is low, as you would expect your isolates

to cluster more closely to each other (unless they were in fact international imports to the hospital).

Line 285: "On the contrary, if there were high transmission across wards, the number of differences when comparing intra versus inter ward isolates would not be different."

Could this not also mean that you have repeat introductions from outside the hospital?

Also - these SNP differences are reasonably high within each ST. Does this not also suggest repeat introductions, rather than a single hospital clone? I have interpreted "independent introductions" to mean a single introduction to the hospital and expansion of the clone from there, but your evidence doesn't seem to support this, so I am confused.

Line 290: "Considering both measures, the distribution of intra ward comparisons overlaps with the distribution of inter ward distribution (see Figure 4)."

I don't understand this, do you have the wrong figure here?

Line 306: Did you look at how gene content correlated with SNP distance? i.e. were closely related genomes based on gene content also fewer SNPs apart?

Additionally, I think more testing and analysis is required for the gene content analysis. Artificial inflation and incorrect assignment of genes caused by annotation errors has been noted previously (see the Panaroo paper, which is the improved tool for pangenome analysis: <https://doi.org/10.1186/s13059-020-02090-4>). I would also take a look at this paper, where they use in silico tests to evaluate the true biological gene variance: doi: 10.1128/mBio.00254-21. Comparing your analysis to something like PopPunk (<http://www.genome.org/cgi/doi/10.1101/gr.241455.118>) which takes into account both core and accessory genome content may also improve the readers confidence in these results.

Discussion:

There is very little discussion or contextualisation with existing literature.

Minor:

line 38: "for a decade" should be "over a decade"

line 73: Use of both "Furthermore" and "also" redundant ("also" is stated again at the end of the sentence - line 75 - suggest removing).

Line 76: The XDR abbreviation needs to be given in line 74

Is Reference 13 truncated? Doesn't include all authors

Line 119: Should specify clinical isolates

Line 130: Remove "was employed" since you have already said "employing" earlier in the sentence

Line 140: "were of high-quality" → "were high-quality"

Line 207: "Please see methods and Supplementary Table 1 for further details about the isolates." ☐
just need "Supplementary table 1" referenced in brackets

Line 219: This sentence is unnecessary: "Table 1 gives the percentage of resistant isolates for each ST for the 6 antibiotics that show some variation across the different STs."

Figures: Check the scales on your trees, some have not rendered correctly

Reviewer comments:

Reviewer #3 (Comments for the Author):

all questions were addressed.

Reply: Thank you for all your comments, they have improved our manuscript.

Reviewer #4 (Comments for the Author):

Mateo-Estrada et al. present a genomic analysis of 73 *Acinetobacter baumannii* isolates collected between 2007-2017 from a tertiary hospital in Mexico. Their main goal was to assess diversity in the hospital over this time and evaluate antimicrobial susceptibility and introduction/transmission of the predominant lineages. One of the most promising aspects of this paper was the use of the accessory genome, which is often overlooked.

Unfortunately, I do not think this study was able to achieve all that it promised in the abstract. There was a general lack of specificity and detail throughout the manuscript, which made it difficult to completely assess the work. I was not convinced by the methods, again potentially due to a lack of information but also because there seemed to be no attempt to validate the results or investigate unexpected outcomes. Conclusions seem to be drawn with very little support or explanation. Generally, the scope of literature referred to in this study appears narrow to me and ignores a large body of work contributed globally by other researchers looking at WGS and *A. baumannii*. I also found the text repetitive and vague in parts and should be made more concise.

Reply: We thank the reviewer for her/his constructive comments. Our manuscript has significantly improved thanks to those comments. After taking into account the comments by 4 reviewers and 2 rounds of revisions, we are sure our manuscript is ready for publication. Of note, the reference to specific lines in the replies below is in the Marked-up copy of the manuscript.

Introduction:

Line 72: should reference the definition of MDR and XDR. Sentence structure would also sound better if said "...due to multidrug resistant (MDR) isolates, defined as resistance to..."

Reply: We have referenced the definitions and changed the sentence structure (see line 74-77).

Line 82: I would suggest the authors look into this publication and cite this instead, as it provides a much more in-depth analysis into the issues surrounding

the *A. baumannii* MLST schemes: <https://doi.org/10.3389/fmicb.2019.00930>

Reply: We know that publication and we have included it. Of note, our work in Emerging Infectious Diseases (ref 9) predates that publication and was the first article to systematically and phylogenomically address the issues of both MLSTs in *A. baumannii*.

Line 84: "WGS has been used to study the spread of clones of *A. baumannii* at national and even at continental levels (9, 10). However, unlike other important bacterial pathogens, WGS has hardly been used to analyse clone diversity within hospital settings. There have been some studies using WGS to study clones of this species within hospitals in some countries, such as UK, Lebanon and Vietnam (11-13). However, very little is known about different lineages causing MDR infections in single hospital settings in Latin America."

I do not agree with some of the statements made here. There are many studies that have made use of WGS to study the global phylogeny of *A. baumannii* and its diversity/transmission within hospital settings. The below is the result of a fairly brief search on the matter:

<https://doi.org/10.1128/AAC.00840-20>
<https://doi.org/10.1099/mgen.0.000052>
<https://doi.org/10.1016/j.ijid.2020.09.123>
<https://doi.org/10.1128/JCM.01818-15>
<https://doi.org/10.1128/mSphere.00934-19>
<https://doi.org/10.1128/AAC.02014-15>
<https://doi.org/10.1073/pnas.1104404108>
<https://doi.org/10.1099/mgen.0.000530>

As such, I think it is untrue to say that WGS has "hardly been used". Even the last statement about knowledge of circulating lineages in Latin America seems somewhat untrue, based on this study from Brazil (<https://doi.org/10.1016/j.ijantimicag.2020.106195>). I would encourage the authors to explore the existing literature more thoroughly in order to reconstruct their introduction to more accurately represent the context of this study.

Reply: We thank the reviewer for the comment. We have changed that the section of the manuscript to better reflect that WGS has been used more frequently than we initially stated (see lines 86-92). We have included the references suggested by the reviewer; of note, one of them was already included in the initial version of the manuscript.

Lines 94-97: please provide references

Reply: References have been provided.

Materials and Methods:

Line 118: The hospital information and isolate information should probably be in a separate section called "study design, settings and isolates". In fact, given that this paragraph is quite long, I would recommend splitting this into separate sections for each (DNA extraction, genome sequencing and susceptibility testing).

Reply: We thank the reviewer for the comment. We have split that paragraph into separate sections.

Line 120: You refer to table 1 here, but this only includes the final 73 isolates, so I think is inappropriate in this sentence.

Reply: We do not refer to table 1 anymore.

Line 122: How many of the original 134 were pulse type 22? Why were these 76 chosen?

It is not clear if there was something more systematic than "trying to represent...the proportion of isolates per year...". Were there any conditions on not having repeat isolates from patients? Having a spread of samples throughout the year? Is there a reason for the large gap in samples from 2012-2015? It may be useful to include in supplementary table 1 the other isolates that were not selected for WGS.

Reply: Only one isolate was taken from each patient. Re the time gap, as mentioned in the methods the isolates came from two previous studies; one from 2007 to 2011 and another covering 2016 and 2017. Unfortunately, we did not get any information for the period 2012-2015. However, this does not mean that there were no *A. baumannii* circulation, we just could not get samples from that period. The 76 isolates were chosen trying to represent as much as possible the proportion of isolates per year. Re the 134 isolates all but three were pulse type 22. All this has been clarified in the methods.

Line 124: I would mention the three isolates being removed in a separate section where you describe how you performed quality control. This would also allow you to mention what exactly it was about the samples that resulted in their exclusion

Reply: Thank you for the suggestion. This has been changed.

Line 131: Please add version numbers for ALL software (this comes up throughout). What qualifies as a "poor-quality" base?

Reply: We have added version numbers for all the programs. Re the "poor-quality" base, we ran Trim galore with the default parameters; thus, the threshold for base quality is a Phred score of 20.

Line 135: In my opinion it is insufficient to just refer to another paper for methods. I think this should be followed up by a brief description of at least the software and versions used. This also allows the creators of the software to be properly credited, rather than re-citing another paper.

Reply: We agree with the reviewer, we have added a brief description and cited the proper programs.

Line 136: Both the Oxford and Pasteur MLST schemes have references, which you can find on the pubMLST website.

Reply: Thanks for the comment, the two references have been added.

Line 155: Given that recombination analysis is one line, it can just be put in this section.

Reply: The recombination analysis was put in the core phylogenies section, as suggested by the reviewer.

Line 169: "we excluded some of the isolates whose genetic divergence and sampling date were unusual."

Which isolates were excluded and why they were "unusual" needs to be described better here, otherwise it just seems like you were fitting your data to suit the model.

Reply: By "unusual" we mean isolates whose sampling date do not seem to be congruent with their genetic divergence. This has been clarified in the text and the names of the excluded isolates have been provided (see line 263-267). Furthermore, Supplementary File 1 provides the list of the isolates used for the molecular dating analysis.

Line 184: Were you looking at core single gene families? Do you have an estimate of the genome size/percent from which you were calling SNPs?

Further to this, I am wondering if your SNP resolution was limited based on this method, and why you chose this method over SNP calling using a single whole genome reference for each ST and read mapping (with Snippy, for example). This would allow for more robust evaluation of each SNP site given the mapping quality and pileup, and would not be affected by possible artefacts introduced with assembly and gene annotation.

Reply: Indeed, we looked at core single gene families. These families represent around 66% of the genome. No, the SNP resolution was not affected by this method. In exploratory analysis we compared SNP calling using a reference

genome and this method and the results of the two methods are in good agreement. We chose this method because it allows also to analyze individual gene families. Furthermore, these core single gene families are extracted from the pangenome analysis, which is also fundamental for the gene content variation analysis.

Results:

Line 210: specify which scheme you mean when saying "belonging to just four STs"

Reply: We have specified the scheme (see line 308).

Line 231: Did you look at the genomic mechanisms for resistance (genes or SNPs)?

Reply: Yes, we did. We conducted an *in silico* prediction of the resistome for these isolates. This would be included in a future study.

Line 238-256:

I have a few comments for this section:

"The phylogeny clearly shows that the STs were introduced in independent events into the hospital."

I would remove this sentence. It's not clear what you mean by "independent events" - it seems like you mean each ST was introduced once and proliferated from there, but you actually go on to describe isolates of the same ST being introduced separately. This sentence also relies heavily on the reader being able to see the phylogenetic tree, which is hard to do since it is in the supplementary.

Reply: We have changed the phrase; now it reads "The phylogeny shows that the STs were introduced separately into the hospital", which we think is more accurate.

Single introduction of ST417 and most of ST208: I am interested to know what further evidence you have that this was indeed a single introduction, and not a lineage in the community or separate healthcare setting causing repeat introductions into the hospital. Were all patients from which these isolates were taken negative on arrival (i.e. all samples were true hospital acquisitions?). Any environmental evidence? This sort of detail may need to be added to the methods section describing the isolates.

Reply: The microbiology laboratory of the hospital classified the samples as coming from hospital infections; because all the isolates came from positive cultures of clinical samples taken from hospitalized patients. In all the cases *A. baumannii* was the etiological agent of the infection and there was no evidence that indicated colonization or community acquired infection in any patient. This information has been added to the methods section (see lines 169-172).

I think you need to spend more time analysing your ST369 and ST136 isolates. It is quite odd in your large tree that you have some ST369 clustering at opposite ends of the tree. Can you explain why? How different is the MLST scheme for ST369 to ST136? The Oxford scheme is notorious for including a capsule gene, which results in different ST assignments due to capsule switches.

Reply: We have spent quite a lot of time on this issue. This issue has to do with recombination affecting the loci used for the Oxford scheme. Furthermore, in a previous paper (see ref 9), we have discussed the issues of polyphyly for ST under the Oxford scheme and, actually, one example was the ST369. In that study we show that 4 of the 7 loci of the Oxford scheme have been affected by recombination. Some of this is mentioned (and was already mentioned in the previous version of the manuscript) in the discussion.

I would say generally that given you have limited geographical context from other South American isolates, support for your "single introduction" events here is low, as you would expect your isolates to cluster more closely to each other (unless they were in fact international imports to the hospital).

Reply: The reviewer is implying that our isolates are from South America but Mexico is not in South America, it is in North America. Just to be clear, we did not infer just one introduction for the all the 73 isolates; actually, 5 introduction events were inferred. Furthermore, just for 2 STs we found a single introduction for each ST. Re the limited context from other genomes, we sequenced as many isolates as we could and downloaded many publicly high-quality genomes from the same STs to provide the broadest geographical context possible. As always, there is only so much you can do given one's funding and the publicly available data. However, we recognized that the data set is not perfect and could be that when more data will be available the number of estimated introductions might change.

Line 285: "On the contrary, if there were high transmission across wards, the number of differences when comparing intra versus inter ward isolates would not be different."

Could this not also mean that you have repeat introductions from outside the hospital?

Also - these SNP differences are reasonably high within each ST. Does this not also suggest repeat introductions, rather than a single hospital clone? I have interpreted "independent introductions" to mean a single introduction to the hospital and expansion of the clone from there, but your evidence doesn't seem to support this, so I am confused.

Reply: This analysis was conducted considering ST208 and ST417. Given our ML phylogenies, whereas for ST417 only one introduction event was inferred, for the ST208 two events were inferred. We also want to mention that, considering time

of introductions inferred by our mol dating analysis, both STs have been evolving in the hospital for several years; thus, it is not unexpected a considerable number of SNP differences within each ST.

Line 290: "Considering both measures, the distribution of intra ward comparisons overlaps with the distribution of inter ward distribution (see Figure 4)."
I don't understand this, do you have the wrong figure here?

Reply: Thank you for noticing this. Indeed, we had the wrong figure there; that has been corrected.

Line 306: Did you look at how gene content correlated with SNP distance? i.e. were closely related genomes based on gene content also fewer SNPs apart?

Additionally, I think more testing and analysis is required for the gene content analysis. Artificial inflation and incorrect assignment of genes caused by annotation errors has been noted previously (see the Panaroo paper, which is the improved tool for pangenome analysis: <https://doi.org/10.1186/s13059-020-02090-4>). I would also take a look at this paper, where they use in silico tests to evaluate the true biological gene variance: doi: 10.1128/mBio.00254-21. Comparing your analysis to something like PopPunk (<http://www.genome.org/cgi/doi/10.1101/gr.241455.118>) which takes into account both core and accessory genome content may also improve the readers confidence in these results.

Reply: This is a nice comment. Yes, we did look at the relationship between gene content variation and SNP accumulation. We published that a few years ago, in 2017 (<https://pubmed.ncbi.nlm.nih.gov/28979253/>); way before both PopPunk and Panaroo were published. In that work we show that gene content variation accrues more frequently and faster than the accumulation of SNPs. In that work we show that pairs of isolates that were identical as per core SNPs could be tell apart by analyzing differences in gene content. Of note, that work was the foundation for the current manuscript and in the introduction and discussion of the current submission we mention this work. We invite the reviewer to read the article.

Discussion:

There is very little discussion or contextualisation with existing literature.

Reply: We have provided contextualization with some of the existing literature (see first paragraph).

Minor:

line 38: "for a decade" should be "over a decade"

Reply: Thank you for the suggestion. We have changed that line (see line 39).

line 73: Use of both "Furthermore" and "also" redundant ("also" is stated again at the end of the sentence - line 75 - suggest removing).

Reply: Thank you for the suggestion. We have removed “also” in line 75.

Line 76: The XDR abbreviation needs to be given in line 76

Reply: We have provided the abbreviation.

Is Reference 13 truncated? Doesn't include all authors

Reply: This has been corrected; now the reference includes all the authors.

Line 119: Should specify clinical isolates

Reply: We have specified that these are clinical isolates (see line 167).

Line 130: Remove "was employed" since you have already said "employing" earlier in the sentence

Reply: We have removed “was employed”, thank you for noticing this.

Line 140: "were of high-quality" ??? "were high-quality"

Reply: Thanks for highlighting this, we have fixed it (see line 224).

Line 207: "Please see methods and Supplementary Table 1 for further details about the isolates." ??? just need "Supplementary table 1" referenced in brackets

Reply: This has been changed as suggested (lines 305-306).

Line 219: This sentence is unnecessary: "Table 1 gives the percentage of resistant isolates for each ST for the 6 antibiotics that show some variation across the different STs."

Reply: The sentence was deleted.

Figures: Check the scales on your trees, some have not rendered correctly

Reply: We have checked the scales on our trees.

June 28, 2021

Dr. Santiago Castillo-Ramírez
Programa de Genómica Evolutiva, Centro de Ciencias Genómicas, Universidad Nacional Autónoma de México
Evolutionary Genomics Research Program
CCG UNAM
Cuernavaca 62210
Mexico

Re: mSystems00626-21R1-A (Accessory genomic epidemiology of co-circulating *Acinetobacter baumannii* clones)

Dear Dr. Santiago Castillo-Ramírez:

Your manuscript has been accepted, and I am forwarding it to the ASM Journals Department for publication. For your reference, ASM Journals' address is given below. Before it can be scheduled for publication, your manuscript will be checked by the mSystems senior production editor, Ellie Ghatineh, to make sure that all elements meet the technical requirements for publication. She will contact you if anything needs to be revised before copyediting and production can begin. Otherwise, you will be notified when your proofs are ready to be viewed.

We recognize that the video files can become quite large, and so to avoid quality loss ASM

suggests sending the video file via <https://www.wetransfer.com/>. When you have a final version of the video and the still ready to share, please send it to Ellie Ghatineh at eghatineh@asmusa.org.

Sincerely,

Jack Gilbert
Editor, mSystems

Journals Department
Supplementary Figure 2: Accept
Supplementary Table 1: Accept
Supplemental Figure 1: Accept
Supplementary Table 2: Accept
Supplementary File 1: Accept